# Forty Years of the Use of Cells for Cartilage Regeneration: The Research Side

**DOI:** 10.3390/pharmaceutics16121622

**Published:** 2024-12-22

**Authors:** Livia Roseti, Carola Cavallo, Giovanna Desando, Martina D’Alessandro, Brunella Grigolo

**Affiliations:** Laboratorio RAMSES, IRCCS Istituto Ortopedico Rizzoli, Via di Barbiano, 1/10, 40136 Bologna, Italy; livia.roseti@ior.it (L.R.); giovanna.desando@ior.it (G.D.); martina.dalessandro@ior.it (M.D.); brunella.grigolo@ior.it (B.G.)

**Keywords:** articular cartilage, transplantation, regeneration, chondrocytes, mesenchymal stem cells, concentrated bone marrow, stromal vascular fraction, secretome, gene therapy

## Abstract

**Background:** The treatment of articular cartilage damage has always represented a problem of considerable practical interest for orthopedics. Over the years, many surgical techniques have been proposed to induce the growth of repairing tissue and limit degeneration. In 1994, the turning point occurred: implanted autologous cells paved the way for a new treatment option based more on regeneration than repair. **Objectives:** This review aims to outline biological and clinical advances, from the use of mature adult chondrocytes to cell-derived products, going through progenitor cells derived from bone marrow or adipose tissue and their concentrates for articular cartilage repair. Moreover, it highlights the relevance of gene therapy as a valuable tool for successfully implementing current regenerative treatments, and overcoming the limitations of the local delivery of growth factors. **Conclusions:** Finally, this review concludes with an outlook on the importance of understanding the role and mechanisms of action of the different cell compounds with a view to implementing personalized treatments.

## 1. Introduction

Human articular cartilage is a hyaline, avascular, aneural, devoid of lymphatics, and low-cellularized connective tissue covering and protecting the ends of long bones in diarthrodial joints [1]. The surface is smooth and slippery with a low coefficient of friction; the deepest layer is calcified (the tidemark) and interlocks with the subchondral bone. Healthy articular cartilage consists of a dense extracellular matrix (ECM) that contains up to 80% water, collagens, and proteoglycans, with a sparse distribution of highly-specialized cells, the chondrocytes. The collagen, mainly type II, forms a network of fibrils that gives the overall shape of the cartilage and provides pockets or compartments filled with the water-binding proteoglycan complexes that regulate the compressibility. Articular cartilage can withstand compression because it contains high concentrations of the polyanionic large proteoglycan aggrecan, trapped within the collagen fibrillar network. The glycosaminoglycan (GAG) chains on the aggregated aggrecan molecules create osmotic pressure, which is held back and contained within the insoluble collagenous meshwork. The chondrocytes balance cartilage homeostasis (ECM synthesis and degradation); they are distributed among different zones, with those that are flat found near the surface, those more profound being round- or oval-shaped, and those in the deepest layers arranged in columns [1,2,3].

Damage to joints is a typical source of pain and disability. Articular cartilage lesions, occurring by trauma, are common in the population and can be chondral and osteochondral [1]. It has been known for years that articular cartilage has a limited healing capacity. The celeb quote by anatomist and physician William Hunter stated, “If we consult the standard chirurgical writers from Hippocrates down to the present age, we shall find that an ulcerated cartilage is universally allowed to be a very troublesome disease; that it admits of a cure with more difficulty than a carious bone; and that when destroyed, it is never recovered.” [4]. 

Numerous surgical techniques address cartilage defects depending on individual patient characteristics and the damage location, size, and depth [5]. However, all concurrent pathologies, including ligament instability, meniscus deficiency, and limb malalignment, must be addressed for cartilage restoration procedures to be successful. Cartilage treatment strategies are palliation, such as chondroplasty and debridement, and repair, such as drilling and microfracture [6,7]. These conventional treatments often lead to the formation of fibrocartilage, which lacks the typical biochemical and biomechanical properties of normal hyaline articular cartilage [8]. 

In 1994, the turning point occurred. For the first time, a Swedish group reported a novel technique using autologous cells to treat cartilage lesions, opening the way to a new concept based on tissue regeneration instead of repair [9]. This procedure, and its later evolution, showed good or excellent results after many years of follow-up [10]. However, due to some drawbacks related to the need for two surgical procedures and the loss of chondrocyte phenotype after in vitro expansion, investigations have been moving toward different cell populations, such as mesenchymal stem/stromal cells (MSCs) [11]. These cells can be mainly isolated from bone marrow or adipose tissue, or artificially generated from terminally-differentiated cells (Induced Pluripotent Stem Cells, iPSCs) [12]. Refinements to the use of cells for cartilage regeneration were represented by what was the emerging field of tissue engineering in the early 1990s [13] This technology relies on the use of a biomaterial scaffold to support cell proliferation, differentiation, and function, to act as a drug carrier, or as a vehicle for biologically-active factors. Further, the use of Bone Marrow Aspirate Concentrate (BMAC) [14] and Stromal Vascular Fraction (SVF) [15] has emerged as an important biological tool for orthopedic surgeons as sources of growth factors exerting anabolic and anti-inflammatory effects, favoring tissue regeneration (Figure 1).

Despite the numerous advances in the field of cartilage regenerative medicine, the identification of the best cell type, the correct number of cells, their fate once implanted in the lesioned area, and the most suitable biomaterial are just a few of the questions still raised by researchers and clinicians [16,17]

The aim of this review is to move from the conception of using autologous cells to the more recent employment of cell-free-derived products, and the likely application of gene therapy for cartilage regeneration, highlighting the biological aspects of the mechanisms underlying their potential clinical translation.

## 2. Cells for Treatment of Cartilage Defects

### 2.1. Chondrocytes

Chondrocytes are the only cell type of cartilage. They are housed in spaces dug into the intercellular substance called *lacunae* that may contain a single or multiple chondrocytes [2]. In the central part of the cartilage, the cells spread in “isogenic” groups, each representing the progeny of a parental cell produced during interstitial growth. The cells are spherical or ovoid. Towards the periphery of the cartilage, the elements become progressively flattened and lose their distribution in groups. Chondrocytes contain a diplosome, numerous mitochondria, elements of granular endoplasmic reticulum, free ribosomes (in which collagen fibrils are synthesized), and an extensive Golgi complex; variable quantities of glycogen particles are also present. Some vacuoles of the Golgi complex contain a finely-granular material; others are filled with electron-dense granules very similar to the granules of the ECM and which stain with Ruthenium Red, a dye for acid GAGs; it is likely that this material present in the Golgi complex encloses proteoglycans elaborated by the ribosomes and the Golgi complex, waiting to be secreted outside the cell. The Golgi complex allows for synthesizing carbohydrates and sulfurizing GAGs, thus constituting the amorphous fundamental substance. The extension of the Golgi complex, the granular endoplasmic reticulum, and the cytoplasmic basophilia increase in the growing or regenerating of cartilage and reduce in the quiescent phase. After being enveloped by the ECM they synthesized, the chondrocytes continue throughout the cartilage growth period to elaborate the matrix and divide (by interstitial growth) [1,2].

A cellular approach for the treatment of cartilage defects started in the 1960s, with many studies using autologous chondrocytes performed on rabbits [18] and, later, sheep [19]. These results encouraged orthopedic surgeons to establish a human treatment model based on autologous chondrocyte implantation (ACI). The first pilot study trial’s results were published in 1994, and the technology subsequently entered the clinic [9]. ACI is a two-step procedure that first foresees a cartilage biopsy harvested from a non-load-bearing area of healthy tissue. This sample allows for a small population of chondrocytes to expand in vitro for a few weeks until it reaches approximately 12 to 48 million cells. The procedure then includes a second operation to implant chondrocytes in the defect site, which is finally covered with a periosteum flap, circumscribing them where necessary and triggering tissue regeneration [9].

Several authors have evaluated the fate of implanted chondrocytes using different experimental animal models. The percentage of the cells engrafted in the lesioned area differed, mainly depending on the species utilized. Some results showed that the cells persisted for at least 14 weeks, providing hyaline-like repair tissue with corresponding improvement in the histologic, biomechanical, and durability characteristics [20,21]. 

Transplanted chondrocytes provide chondrogenic factors selectively stimulating chondrogenic processes, such as precursor cell and chondrocyte proliferation and the synthesis of ECM components like type II collagen and proteoglycans. Several studies have evaluated their role in molecular signaling and subsequent tissue remodeling, leading to mature hyaline-like cartilage formation. A disadvantage of chondrocyte use is the limited number of cells after isolation and possible dedifferentiation during the expansion phase, suggesting the importance of the microenvironment for cell growth and phenotype preservation [22].

### 2.2. Tissue-Engineered Chondrocytes 

Since the first report by Brittberg et al. on the first 23 patients [9], more than 30,000 patients have undergone ACI, globally. Clinical results have been reported from various centers around with a high percentage (84-90%) of good-to-excellent clinical outcomes, especially in patients with various types of femoral monocondylar lesions [23,24,25]. 

However, despite very promising results with ACI, the potential for calcification associated with large surgical incisions, peripheral graft enlargement, graft dissection, and the use of periosteal flaps, have hindered the effectiveness of this treatment strategy. Postoperative experience indicates that a significant percentage of patients (20%) develop symptoms of joint traction due to enlargement of the edge of the periosteal graft, requiring revision arthroscopic surgery. Poor integration of the periosteal flap and periosteal avulsion have been reported, confirming that the periosteum may form ectopic bone [26,27]. Complications arising from the periosteal flap have prompted the search for alternative solutions, such as using different materials and type I/III collagen membranes [28,29].

In addition to the problems mentioned above, chondrocytes grown in monolayer cultures lose their differentiated phenotype, acquiring a fibroblastic-like structure. This process can be prevented by growing the cells in a 3D environment that allows them to re-express cartilage-specific genes, as already demonstrated by Benja et al. in 1982 [30]. The in vitro conditions are very different from those of the human joint, a complex heterogeneous environment whose balance is modulated by different biological and mechanical stimuli [31].

Besides maintaining the chondrocyte phenotype, some scaffolds participate in cartilage regeneration due to their active and informative role [28,29]. To this end, our group demonstrated in a preclinical study the ability of a hyaluronan-based scaffold to play an instructive role by down-regulating the factors involved in cartilage degenerative diseases, suggesting its use to treat early lesions in OA patients also [32].

Autologous chondrocyte matrix-induced chondrocyte implantation (MACI^®^) uses a type I/III collagen membrane and Tisseel^®^ fibrin sealant to anchor the chondrocyte-seeded scaffold and fill the void in the cartilage defect [33,34]. Results showed improvements in clinical and functional parameters; MRI confirmed the presence of hyaline-like cartilage and no complications [33]. In addition, the development of an arthroscopic implantation technique has represented a significant improvement in the transplantation strategy. It is beneficial in treating large defects because it avoids donor site morbidity while maintaining the advantages of the arthroscopic osteochondral transplantation technique [35].

Over the last years, several natural or synthetic biomaterials have been tested and utilized for cartilage tissue engineering [36], to find the best solution that fulfils the critical requirements such as biocompatibility, biodegradability, permeability, porosity, nutrients and waste diffusion, and mechanical properties.

Recent progress in the field of biomaterials has opened promising prospects for the use of injectable hydrogels. This versatile class of scaffolds, which can mimic the properties of natural articular cartilage, has undeniable advertising advantages with minimally-invasive effects on damaged areas [37,38].

Injectable hydrogels can serve as delivery vehicles for cells and biological molecules. The development of novel drugs in combination with a suitable release carrier is a promising approach to obtaining efficacious treatments for cartilage lesions or diseases.

### 2.3. Mesenchymal Stem Cells

Despite favorable clinical results, ACI still displays issues, like the scarce availability of the cell source, donor-site morbidity, and the switch from a chondrocyte to a fibroblast phenotype during monolayer culturing [39]. These drawbacks have led to the research of other cell types, like MSCs, which are adult multipotent progenitors widely located in specialized niches of most tissues. They can migrate toward the sites of injury, inflammation, and tumors, playing a key role in tissue repair and homeostasis [11], even if their ability and number diminish with age [40].

Bone marrow (BM), adipose tissue (AT), and, recently, synovium (membrane, SM, and Fluid, SF) are the most significant sources of MSCs for cartilage regeneration [41]

MSCs were discovered within the BM stroma. However, rather than a mere supporting function, the MSCs’ in vitro stemness was demonstrated [42], as these cells showed self-renewal and formed fibroblastic colony-forming cell units. MSCs were also proven to be multipotent, differentiating first into three cell lineages: adipocytes, osteoblasts, and chondrocytes [43], and then into many other cell types [44].

Since MSC cultures are heterogeneous, no unique marker is available for their identification. The International Society of Cellular Therapy (ISCT) recommended that MSCs must meet minimum criteria, including adhesion to plastic, expression of surface markers CD73, CD90, and CD105 (≥95% expression), the absence of hematopoietic markers CD34, CD45, CD14, or CD11b, CD79α, or CD19 (≤2%), the lack of HLA Class II molecules, and trilineage differentiation, chondrogenic, osteogenic, and adipogenic [45,46].

Other significant discoveries about MSC regenerative potential were their anti-fibrotic, anti-apoptotic, anti-inflammatory, pro-angiogenic, immunomodulatory, and chemoattractive effects. MSCs’ immunosuppressive capabilities have also opened the way for their use in allogeneic treatments. However, attention should also be paid since such capabilities can be converted under specific conditions into immunostimulating ones [47].

Compared to chondrocytes, MSCs can expand to higher cell numbers in culture [41]. Monolayer cultures and pellets/high-density aggregates/mass are significative culture models developed to study in vitro chondrogenic differentiation, eventually stimulated with pro-chondrogenic medium (serum-free DMEM, TGFβ3, dexamethasone, and insulin), allowing for the mimicking of the chondrogenic process of embryogenesis [48]. During this process, cartilage acts transiently as a template for bone; chondrocytes become hypertrophic by depositing a collagen-type X-rich matrix, and undergo apoptosis to allow osteoblasts to infiltrate, depositing a mineralized bone matrix [49]. In vitro, chondrogenesis occurs but happens parallel to the differentiation process during fracture callus wound healing [50]. Other in vitro conditions are 2D and 3D cocultures with chondrocytes, the combination with biomaterials/scaffolds which has been known for years and is still being investigated [51], or the control of specific parameters like oxygen tension [52] or mechanical stimuli (bioreactors) [53].

In the 2000s, the hypothesis was that MSCs were identified as pericytes, perivascular cells capable of migrating to damaged areas for healing, secreting several molecules that exert trophic and immunomodulatory effects contributing to tissue repair. However, some studies do not support this hypothesis [54]. Subsequent discoveries highlighted that the paracrine effects are produced by MSCs at injury sites by secreting soluble molecules like growth factors, cytokines, and chemokines. This paracrine activity has led to the proposition of an additional designation for these cells under the MSC acronym “Medicinal Signaling Cells” [55]. The following studies indicated that MSCs secrete growth factors, cytokines, and other bioactive factors throughout exosomes and microvesicles [56].

AT stem/stromal cells (ASCs) or adipose-derived stem/stromal cells (ADSCs) were first characterized in 2001 as deriving from the SVF of lipoaspirate (the product of liposuction) [57].

ASCs display several features of BM-MSCs, like phenotype, the ability to release chondrogenic, anti-fibrotic, and anti-apoptotic factors, and a role in suppressing immunoreactions [58]. Those characteristics make ASCS valuable for regeneration but with some advantages: AT is more accessible and has a cell density higher than BM; the harvesting procedure is less invasive; and ASC proliferative capacity is more prominent than the BM-MSC one [59,60].

Among the investigations on ASCs that have shown encouraging results for cartilage regeneration, our study in a rabbit OA model also revealed a capacity for attenuating inflammatory events in the synovial membrane, inhibiting OA progression [61].

Recent studies have shown that synovium is a rich “niche” for MSCs. SM-MSCs have higher colony formation efficiency and chondrogenic and proliferative potential than donor-matched BM-MSCs and AD-MSCs [62]. They are less affected by donor age than AD-MSCs [63]. Transplanted SM-MSCs differentiated into cartilage in rabbit articular defects [64] or as an MS MSC-ECM-based construct in pigs [65].

Synovial fluid MSCs (SF-MSCs) hold promise as a source of cells for cartilage regenerative therapies: they express CD44, CD49c, CD49f, CD151, CD140, and CD146 and can undergo chondrogenic differentiation [62]. They may derive from various joint tissues, but the exact origin is unclear [66].

Despite the wide clinical use of MSCs, safety issues are still a matter of debate, especially in the long-term follow-up. The main concern is the possibility of unwanted differentiation of transplanted MSCs and the generation of new blood vessels that may inhibit the anti-tumor immune response and promote mutagenesis and cancerogenesis [67,68].

Basic biological research is necessary to understand the mechanisms of action and signaling pathways that determine the fate of MSCs after administration.

### 2.4. Tissue-Engineered Mesenchymal Stem Cells

Like chondrocytes, MSCs need a vehicle to hold them at the site of lesion/damage for cartilage regeneration, allow their growth, and protect them from unfavorable environmental conditions. Natural scaffolds, such as fibrin and collagen, show good biocompatibility but have limitations in fabrication, mesh properties, biodegradability, and risk of immunological reactions. Synthetic scaffolds offer better versatility and controllable biodegradability but may pose cytocompatibility issues and toxic by-products. Developing composite, zonal, and nanofibrous scaffolds aims to address these limitations [59].

Our group evaluated the ability of BM-MSCs to differentiate in the chondrogenic sense when seeded onto an HA-based scaffold, previously utilized in chondrocyte research and clinics [69] and implanted in a rabbit OA condyle. The results demonstrated that the cells expressed cartilage-specific markers [70]. Moreover, we found that a scaffold composed of type I collagen and magnesium-enriched hydroxyapatite and zonally organized to mimic the native osteochondral structure, could support the growth and differentiation of expanded human MSCs toward a chondrogenic or osteogenic phenotype, depending on the layer [71].

A recent study evaluated intra-articular injections of ASCs seeded in an amnion membrane (AM)-based hydrogel in a collagenase-induced OA rat model. AMs and ASCs, rich sources of bioactive molecules, acted synergistically, showing a regenerative and anti-inflammatory effect [72].

Lastly, the use of SM-MSCs combined with scaffolds, such as chitosan hydrogel/3D-printed poly(ε-caprolactone) [73], polydopamine [74], or cartilage ECM-based scaffolds, has been shown to promote cartilage formation in animal models [75].

## 3. Concentrated Cells for Treatments of Cartilage Defects

### 3.1. Bone Marrow Concentrate

Despite good clinical results, ACI remains at a significant disadvantage due to the need for two surgeries and its high cost [76]. The limited donor space available, the risk of damage to healthy joints during the plug collection phase, the discontinuity of cartilage repair obtained at the recipient site due to slightly different plug orientation, and the need for open surgery at the donor and recipient sites are major disadvantages [76]. Therefore, new cartilage repair methods were developed to overcome these shortcomings.

The use of MSCs as pure cell lines after selective removal of cells that do not express typical markers of MSCs has been advocated [77,78,79]. However, the basis of the described technique lies in the critical role of the surrounding microenvironment (or niche). The potential of pluripotent cells considers not only the intrinsic capacities of the cell itself but also the interaction of the cell with physiological niches that provide signaling networks (i.e., ECM, adhesion molecules, growth factors, cytokines, and chemokines secreted by resident cells). Autologous BM contains not only stem and progenitor cells, the source of regenerative tissue, but also accessory cells that support angiogenesis and angiogenesis by producing various growth factors [80].

This suggests that cell selection and expansion in the laboratory, which are highly time-consuming and cost-intensive procedures that necessitate Good Manufacturing Practice (GMP) facilities (as for ACI or MSC transplantation), are not required.

However, BM Aspirate (BMA) has been rarely used, since only about 0.001% of nucleated cells are MSCs [81]. Therefore, attempts have been made to increase their number through concentration, usually by density gradient centrifugation. This concept of concentrating BMA to produce BMA Concentrate (BMAC) allows for increasing not only the numbers of MSCs, but also platelets containing growth factors and hematopoietic stem cells (HSCs) per sample volume. The platelet component of BMAC releases growth factors that initiate the migration of stem cells to the site of injury and provide adhesion sites for migrating stem cells [82]. In addition, HSCs differentiate into blood cells, supporting the vasculature and maintaining cell-to-cell contact with MSCs that stimulate both chondrogenesis and osteogenesis [83]. BMAC is one of the few stem cell transduction methods approved by the US Food and Drug Administration (FDA) [84]. Several commercial systems are available from various manufacturers, but it is difficult to choose the best one as each includes a centrifuge and does not provide equivalent cell counts. Standardizations of all these factors are needed, as they may provide a more scientific way of evaluating possible effects. This is especially important because large individual differences exist in the number of growth factors produced and secreted by the different cell populations within the BMA [84].

### 3.2. Tissue-Engineered Bone Marrow Concentrate

Biomaterials such as scaffolds have potential applications in tissue regeneration, but their application in regenerative medicine is limited due to poor cell infiltration, proliferation, and differentiation within these structures [14]. To address these issues, adding BMACs to biomaterial scaffolds and gels has been highlighted as a possible solution to provide localized growth factors that promote cell infiltration, proliferation, and differentiation [85]. Incorporating BMACs into biomaterials can increase the levels of pluripotent cells that differentiate into local cell types and improve the health of the target site. The most common biomaterials used in clinical practice for PRP and BMAC are the natural biomaterials collagen, hyaluronic acid, and the synthetic biomaterial polylactic-co-glycolic acid. Biomaterials designed to mimic local, healthy signaling pathways and intrinsic mechanical properties can be incorporated into local tissue structures, minimizing host–biomaterial interactions. Biomaterials can, therefore, be applied not only as therapeutic carriers for PRP and BMAC, but also as functional regenerative scaffolds for cell integration, proliferation, and differentiation that can promote musculoskeletal tissue healing on a macro scale [14].

### 3.3. Stromal Vascular Fraction

SVF, derived from AT, is a heterogeneous mixture including ASCs, fibroblasts, erythrocytes, endothelial cells, residual ECM, and soluble factors such as growth factors, adhesion molecules, cytokines, and chemokines [86]. Like BMC, SVF has its regenerative potential in its “niche” containing ASCs and their microenvironment; moreover, it bypasses the costly and time-consuming expansion phase typically required in cell cultures [87]. Unlike ASCs implied in allogeneic and autologous treatments, SVF is only suitable for autologous therapies due to various cell types that may cause immunological rejection.

The characterization of SVF’s regenerative cells has yet to reach a broad consensus. The ISCT and the International Federation for Adipose Therapeutics and Science (IFATS) updated the definition of SVF cells based on surface antigens. They have identified the CD34 antigen as an SVF-ASC marker crucial for regeneration, even though it is primarily associated with hematopoietic stem cells [15].

Two primary methods for obtaining SVF from AT are enzymatic digestion, which breaks down the ECM and releases the SVF cells, and non-enzymatic isolation, which relies on mechanical techniques such as centrifugation [88].

AT samples of mechanical micro-fragmentation (MF), enzymatic digestion (SVF), or manipulation for cell isolation and expansion (ASCs) were investigated in vitro and in vivo in an OA rabbit model. All samples yielded 85–95% of viable cells. In vivo, no significant side effects or inflammatory responses were observed. MF showed the best results in qualitative and semi-quantitative evaluations of articular cartilage [89].

Furthermore, while initial studies primarily focused on lipoaspirate as a source of SVF, the discovery of the knee’s infrapatellar fat pad (IFP), also known as Hoffa’s body, as a new source of multipotent stromal cells, has opened new avenues for research and therapeutic applications [90].

### 3.4. Tissue-Engineered Stromal Vascular Fraction

The use of scaffolds offers opportunities to improve SVF applications.

AT-SVF combined with a microfracture–hyaluronic acid (MF-HA) scaffold was more effective than using an MF-HA scaffold alone for enhancing osteochondral regeneration. Consequently, this combination can be employed alongside microfracture and scaffolding techniques to promote cartilage regeneration, especially in secondary OA [91].

SVF or cultured ASCs associated with collagen I/III scaffolds induced cartilage and subchondral bone formation in osteochondral knee defects of a goat model. The presence of other cell types that may exert synergistic effects at later stages and SVF’s higher differentiation potential led to better results after SVF treatment [92].

## 4. Induced Pluripotent Stem Cells

iPSCs have become a powerful tool in basic, translational, and clinical applications because they can be maintained indefinitely while preserving the host’s genetic makeup.

An iPSC is a type of stem cell artificially generated from a terminally-differentiated one (usually an adult somatic cell) through the introduction of four genes, specific ones encoding some transcription factors that induce their conversion into a stem cell of a specific cell line, which, in turn, can develop into a differentiated cell [12].

Various reprogramming approaches have been developed, which include viral and non-viral vectors and the recent chemical technique. Viral vector integration into the genome may result in insertional mutagenesis and undesired transgene reactivation. It is also important to address the drawbacks of alternative methods. iPSCs allow for overcoming limitations associated with adult somatic cells since possessing self-renewal ability, high developmental plasticity, and immunogenic properties without issues correlated to invasive cell harvesting or loss of mitogenic activity in vitro [12].

Several preclinical models have been developed to drive the differentiation of iPSCs to the chondrogenic lineage, like high-density pellet culture systems enriched with TGF-β or Bone Morphogenetic Proteins (BMPs). However, the resulting cartilage generally combines hypertrophic, articular, and fibrous tissue. That feature can be avoided by differentiating MSCs from iPSCs, as performed by Guzzo et al. [93]. They found that using both the parental precursor iPSCs and the iPSC-derived MSC-like (iPSC–MSC) population, together with an optimized high-density culture environment, the chondrogenic potential of iPSCs was enhanced, and the heterogeneity of the differentiated progeny limited [93]. The main safety issue in iPSC-based treatment is the risk of teratoma formation that can occur if the patient receives iPSC-derived cells containing undifferentiated iPSC. After transplantation, iPSCs’ non-controlled proliferation and differentiation can result in tumor generation or undesirable differentiation [94]. Therefore, the development of more effective methods for the generation of purified populations of autologous iPSC-derived differentiated cells remains a challenge for personalized regenerative medicine [95].

## 5. Transitioning Towards Cell-Derived Products: Secretome

Recent studies have shown that MSCs’ therapeutic potential in tissue regeneration lies in their paracrine effects, which play an important role in regulating fundamental biological processes thanks to the many bioactive factors produced by MSCs [96]. This set of secreted factors/molecules is named “secretome”. It includes GFs, cytokines, chemokines, soluble proteins, nucleic acids, lipids, and extracellular vesicles, all involved in tissue regeneration [97] (Figure 2).

Various groups have reported MSC secretome composition characteristics [98,99], varying according to cell source, culture medium, and incubation time. MSCs secrete several GFs involved in cross-talk with chondrocytes, including insulin-like GF-1 (IGF-1) [100], transforming GF (TGF) [101], and bone morphogenetic protein-2 (BMP-2) [102], which are involved in several processes related to chondrocyte anabolism and cartilage matrix repair. In addition to GFs, MSCs secrete other factors, like, for example, thrombospondin-2 (TSP-2), identified in the secretome of umbilical cord blood-MSCs, that can stimulate chondrogenic differentiation of chondrogenic progenitor cells in vitro and in vivo and promote cartilage regeneration [103]. A proteomic analysis comparing the secretions of ASCs, BM-MSCs, and SM-MSCs exposed to IL-1β and TNF-α observed that inflammatory stimuli promote the production of paracrine factors involved in cartilage repair. For example, MSCs’ secreted proteins are associated with the TGF-β signaling pathway, involved in collagen and aggrecan expression [104].

The secretome’s ability to facilitate cell-to-cell communication and influence their biological functions has attracted the interest of researchers who could use the secretome for cell-free therapy. The secretome can help overcome some problems associated with the transplantation of living, proliferating cells, such as tumorigenicity, immunogenicity, and the risk of infectious disease transmission [105].

Paracrine factors present in CM can reduce catabolic processes in damaged cartilage, such as matrix degradation by MMPs and the production of inflammatory cytokines.

At the same time, CM promotes anabolic processes and induces an anti-inflammatory environment by increasing the expression of genes involved in ECM synthesis. As a result, CM corrects the imbalance that causes cartilage degradation due to the inflammatory scenario seen in patients suffering from degenerative cartilage disease.

Many in vitro studies explored the MSCs’ secretome. To clarify the effects of those soluble factors, secretions of MSCs from different origins were tested for cartilage repair. Different animal models of cartilage repair have confirmed the chondroprotective effects of MSCs-CM treatment in vivo, as shown in vitro. In an OA rat model, a study compared the effects of ASC-CM and shockwave (SW) therapy, a non-invasive approach to promote cartilage regeneration. Results showed that CM treatment restored hyaline cartilage, diminished inflammatory factors, and increased cartilage genes like SOX9 and COL2. This effect was dose-dependent, as observed when SW was administered [106].

However, despite encouraging evidence of the secretome’s beneficial effects in musculoskeletal disorders, its therapeutic use should be subject to regulatory requirements for manufacturing and quality control to establish the product’s safety and efficacy profile [107].

An ideal secretome for cartilage regeneration (whole or EV only) should be able to modulate the immune response, reduce inflammation at the lesion site, lower metalloproteinase levels, and increase activation of resident chondrocytes to produce new ECM. Hydrogel systems could be engineered to promote MSC proliferation and maintain regenerative properties during ex vivo expansion, improve MSC survival, retention, and engraftment in vivo, and direct the MSC secretory profile using tailored biochemical and biophysical cues. A biocompatible and biodegradable hydrogel will efficiently dissolve in the surrounding tissue and deliver the secretome in a controlled manner, optimizing its function. The knowledge accumulated over the last few years, together with advances in the isolation and characterization of secretome through cell culture and hydrogel production, shows that there is still a need to improve the relationship between the secretome and hydrogels [108].

## 6. Gene Therapy for Treatments of Cartilage Defects

Unlike bone, cartilage cannot build an appropriate repair response due to its hypocellularity and complete absence of vasculature or innervation [109]. In general, gene therapy for cartilage aims to reduce joint inflammation, inhibit ECM degradation, and increase cartilage synthesis [110].

Gene therapy refers to delivering nucleic acids to target tissues using viral or non-viral vectors to prevent, treat, or cure disease [111]. There are two primary ways to administer gene therapy: ex vivo or indirect delivery, where cells are harvested from the patient, exposed to a vector, and reinserted into the patient’s joint; and in vivo or direct delivery, which involves injecting the genetic material directly into the joint [112] (Figure 3).

Orthopedic applications of this therapeutic approach have, in particular, the advantage of local delivery to individual locations, such as joints or sites of tissue injury, which significantly reduces the required amount of vector, improves safety, and lowers costs [110,111,112].

The genes employed in cartilage gene therapy that showed a potential for healing are growth factors, anti-inflammatory molecules, therapeutic genes (aggrecan, collagen type II, and SOX-9), and non-coding RNAs [113].

Allogeneic chondrocytes were transduced in vitro with adenovirus vectors expressing insulin-like growth factor-1 (IGF-1) [114] or bone morphogenetic protein-7 (BMP-7) [114]. In both cases, gene transfer accelerated early healing but at later time points. Only when AAV was used to deliver IGF-1 into autologous chondrocytes did a longer-lasting improvement occur [115].

A transfection experiment of lipid-mediated plasmid DNA carrying the gene TGF-β 1 (pDNA-TGF-β1) in MSCs sown on a hybrid scaffold of PLGA/fibrin gel showed good regeneration of cartilaginous tissue [116]

ASCs transfected with nanoparticles of dexamethasone-conjugated polyethylenimine (DEX PEI) complexed with a minicircle plasmid (MC) carrying the SOX-9, 6 and small hairpin RNA targeting ANGPTL4 (shANG)—MC SOX9/6/shANG-tADSCs—showed significantly higher expression of the COL2 gene and protein compared to those transfected with MCSOX9/6 (MC SOX9/6-tADSCs) during in vitro chondrogenesis. Both MC SOX9/6/shANG-tADSCs and MC SOX9/6-tADSCs enhanced chondrogenesis, even without the addition of growth factors, in comparison to negative controls [117]. A vector within a scaffold might offer advantages by increasing the gene delivery rate into the cells in contact with the gene-eluting scaffold and allowing gene spatiotemporal control [118].

Despite its potential, gene therapy still poses challenges. The first issue is safety: viral vectors’ immune activation and tumorigenic side effects still demand attention. Using non-integrated and low-immunogenic rAAV or safer non-viral vectors may be a more promising direction [119]. Second, the functions of genes are diverse and can lead to unpredictable, harmful pathological processes. For example, RelA/p65 is a potent transcriptional activator of ADAMTS5 in chondrocytes, but it can also induce anti-apoptotic genes to protect chondrocytes from apoptosis during the development of OA [120]. In articular cartilage, heterozygous knockout of RelA caused a considerable acceleration of OA by increasing chondrocyte death despite reduced expression of catabolic genes [121]. Using multigene regulation instead of a single gene may be more practical. Third, gene expression requires dynamic homeostasis, or unintended adverse effects may occur. For example, excessive TGF-β in OA can lead to synovial fibrosis, osteophyte formation, and subchondral bone changes [122]. Potent overexpression of TGFβ1 in OA rabbit joints damaged cartilage ECM [123]. The above-mentioned self-feedback gene regulation circuit using CRISPR/Cas9 technology has an environment-dependent feature and can automatically regulate gene expression on demand. The recently developed CRISPR/Cas9 technology provides a simple and efficient option for gene editing [124].

The application of gene therapy to regenerative orthopedics has generated much interest and many publications, but clinical translation still needs to be improved. Invossa, an ex vivo gene therapeutic (TissueGene-C; TissueGene Inc., Rockville, MD, USA), received approval in South Korea in 2017, but it was then retracted in 2019, remaining under appeal; a Phase-III Invossa clinical trial restarted in the USA.

The extent to which clinicians use gene therapy depends not only on its safety and effectiveness, but also on its cost. Currently, approved gene therapy methods are very expensive since they are systemically administered in large amounts. In contrast, most clinical applications in orthopedics involve small quantities of joint injections.

The repair of cartilage injuries using gene therapy will gain increasing importance shortly. The association of gene therapy and tissue engineering represents an important advancement in this field.

## 7. Discussion

Many years have passed since the first application of cells to repair cartilage lesions, a treatment that completely revolutionized orthopedics’ approach to injuries affecting this tissue.

Such a new strategy has paved the way for a regenerative rather than a reparative approach. It changed the concept of healing tissues and organs by restoring their function lost due to aging, disease, damage, or defects.

Moreover, this therapeutical strategy filled the gap between the use of physical/pharmaceutical and traditional surgical techniques and prostheses, providing a more biological approach favorably accepted by patients of every age.

Autologous chondrocytes were the first cell population used for cartilage regeneration. However, despite the success of their clinical application, some drawbacks related to the surgical procedure and biological cell activity prompted the search for new therapeutic solutions, moving from the use of mature or progenitor cells to concentrates, and, recently, to cell-derived products.

Tissue engineering was developed to resolve some of the problems related to in vitro expansion by combining cells with biomaterials and growth factors. In 1993, Vacanti published a relevant paper in *Science* entitled “Tissue Engineering,” where he defined tissue engineering as “an interdisciplinary field that applies the principles of engineering and the life sciences toward the development of biological substitutes that restore, maintain, or improve tissue function” [13].

The past decades have been devoted to regenerative medicine, in which scientists, engineers, and physicians collaborated to construct biological substitutes for regenerative purposes.

Different tissues and cell types have been evaluated to find the ideal source. Each cell population has been shown to possess specific characteristics that drive the processes leading to tissue neoformation, which is positively addressed by the triggering role of biological scaffolds and growth factors.

The use of adult or progenitor cells for regenerative medicine allowed to highlight their therapeutical performance and, importantly, to evaluate in depth the biological pathways responsible for their specific activity. This has led to the use of different “orthobiologics” for different purposes, extending their application to musculoskeletal pathologies such as osteoarthritis (OA) and other cartilage diseases [12].

The need for an easy-step procedure has led to the idea of using concentrated BM or AT, which contain not only cells with regenerative potential, but also cells that support angiogenesis and vasculogenesis by producing several growth factors suitable for osteochondral lesion treatment [125].

Recently, intensive research on stem cell therapy has shown that the benefits of these cells are due to their paracrine action. Direct delivery of stem cell “secretome” to the site of injury demonstrated comparative therapeutic effectiveness of cells while avoiding potential limitations due to their manipulation, even if minimal.

Gene therapy could solve some of the issues of cell therapeutic procedures used for cartilage regeneration. This approach could represent a valuable tool to successfully implement this kind of treatment, bypassing the issues due to growth factor local delivery, such as the short half-life, large dose requirement, high costs, need for repeated applications, and poor distribution [118].

The identification of additional molecular targets and the development of new gene delivery techniques will improve clinical translation also when musculoskeletal pathologies present a genetic etiopathogenesis.

Even if all the procedures have pros and cons in their application and clinical results, they represent innovative and evolving solutions for cartilage tissue regeneration.

Moreover, the huge number of studies on this topic enlarged our knowledge about cells of both human and animal origin, highlighting their mechanisms of action and the principal pathways involved in cartilage regeneration.

However, the lack of a standardized system for describing cell therapies has hindered the clinical practice. In 2019, a group of experts published a manuscript reporting an international consensus on strategies to improve standardization and transparency when describing these therapies. The use of this tool could improve clinicians’ and patients’ abilities to discriminate between the different product preparations [126].

As reported by Brittberg in April 2024, young donor allogeneic cell sources for chondrocytes and MSCs with large-scale productions, ensuring a stable chondrogenic quality, will probably become the future option [10].

We believe that, in the next years, the evidence of the role and mechanisms of actions exerted by the different “orthobiologic” compounds will grow and will provide a more

solid scientific rationale at the basis of their application.

Moreover, the use of cell-derived products such as exosomes, with their anti-inflammatory, anti-apoptotic effects, as well as regeneration by macrophage polarization, will represent a powerful tool for the future treatment of cartilage diseases, creating new opportunities for regenerative medicine strategies. This will simplify the clinical procedures and the possible utilization of allogenic donors reducing, in the meantime, the associated costs.

Although the number of clinical studies is still limited, there is evidence that the beneficial effects of these products can be further enhanced through bioengineering, gene therapy, biophysical and biochemical stimulation, and drug encapsulation.

## Figures and Tables

**Figure 1 pharmaceutics-16-01622-f001:**
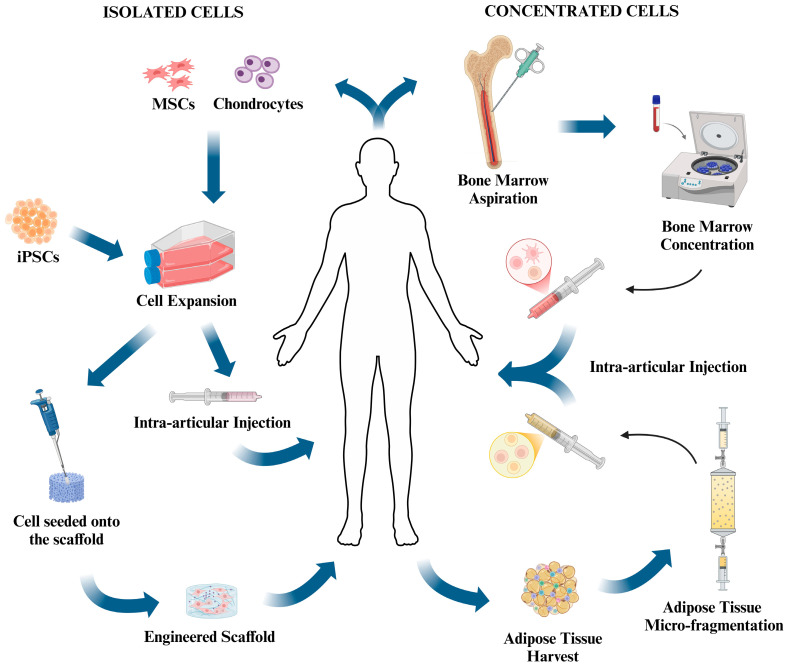
Image of the therapeutical options for cartilage regeneration. In the left side of the image, from top to bottom, the process of cell therapy starts with chondrocytes or MSCs isolated from the source tissue or IPSCs; the process continues with cell expansion in culture and ends with the resulting cell suspension injected into the patient’s lesion. In the left side of the image, at the bottom, the process of tissue engineering where, after expansion, the cells are sown and cultured on a scaffold of various origin and composition, the engineered scaffold is then implanted back into the patient. The right part of the image depicts the production processes of bone marrow and adipose concentrate, from top to bottom: collection of bone marrow from the iliac crest of the patient, production of the bone marrow concentrate by centrifugation, and intra-articular injection back in the patient; from bottom to top: fat tissue harvest from the patient, production of the adipose concentrate by micro-fragmentation and intra-articular injection back in the patient. The image was created with BioRender.com.

**Figure 2 pharmaceutics-16-01622-f002:**
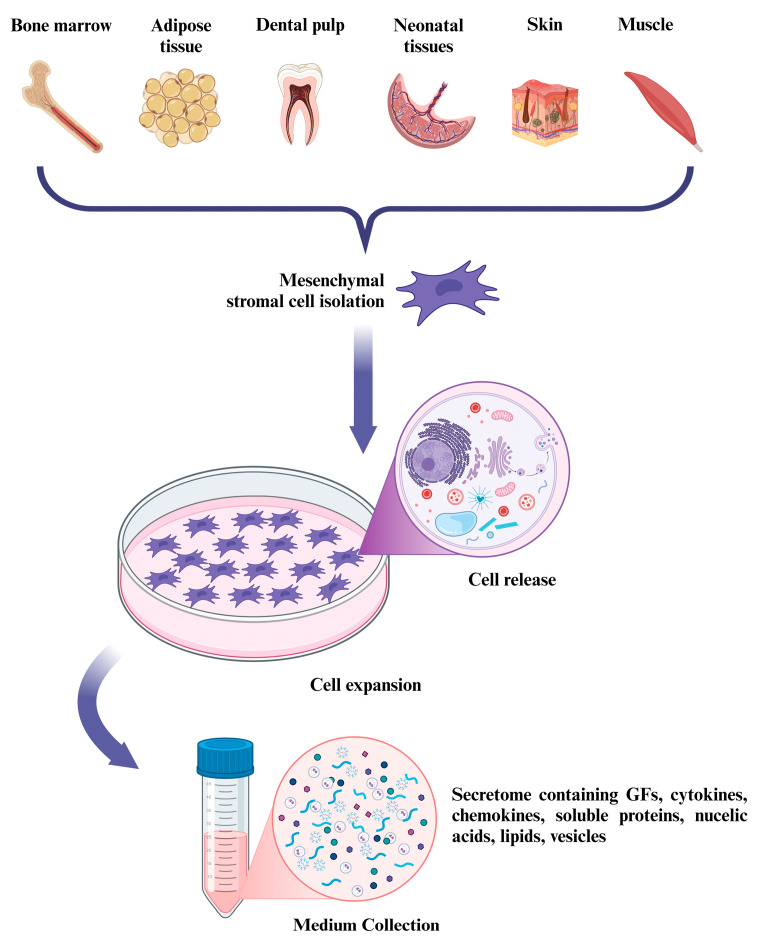
MSC-derived secretome production. From top to bottom: MSCs are isolated from different tissue sources; expanded MSCs are modified with therapeutic molecules (i.e., mRNA, miRNAs, and cytokines), cultivated under defined conditions (serum deprivation, hypoxia), or chemically or physically stimulated; the set of secreted factors/molecules, named “secretome”, is isolated from MSC-conditioned media, purified, characterized, and quantified. The image was created with BioRender.com.

**Figure 3 pharmaceutics-16-01622-f003:**
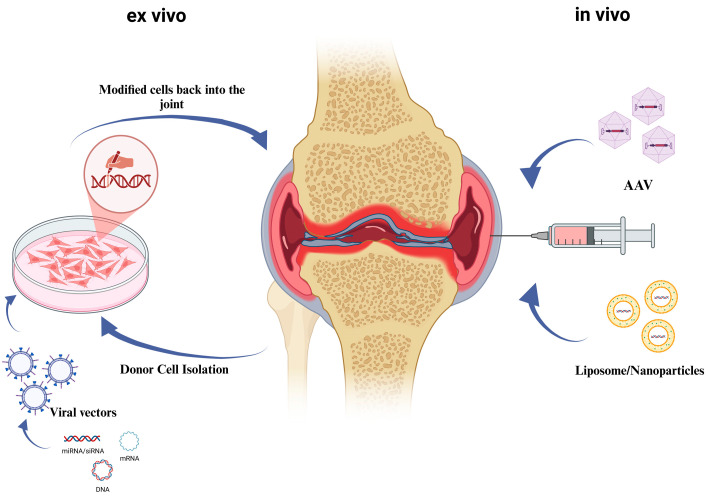
Schematic representation of ex vivo and in vivo workflow for gene therapy administering into joints for cartilage regeneration. Gene therapy approaches are as follows: ex vivo (left side, from down to top): genetic modification and creation of a virus-based product containing the modified gene; cell isolation from the donor and culturing; cell transfection in culture; transferring of the transduced cells back into the joint of the patient. In vivo (right side, from down to top): direct delivery to the joint of the patient using viral (for example, Adeno-Associated Virus, AAV) or non-viral (liposome, nanoparticles) vehicles carrying a therapeutic transgene. The image was created with BioRender.com.

## Data Availability

Not applicable.

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
