# Peer review of "Forty Years of the Use of Cells for Cartilage Regeneration: The Research Side"

_pharmaceutics, 2024, doi:10.3390/pharmaceutics16121622_

Round 1

Reviewer 1 Report

Comments and Suggestions for Authors

This manuscript provides an interesting summary of the development of cell-based therapies for cartilage regeneration. However, several issues need to be addressed. 

The text includes statements such as, “In 1994, the turning point. For the first time… (line 56),” “A cellular approach for the treatment of cartilage defects started in the 1960s… (line 89),” “The results of the first pilot study trials were published in 1994… (line 93),” and “The past three decades have been devoted to regenerative medicine… (line 471).” However, the claim of “Forty years” in the title lacks clarity and precision, which is important for understanding the work.

Additionally, the inclusion of induced pluripotent stem cells (iPSCs) in the section on mesenchymal stem cells (MSCs) is not well justified. Creating a separate section for iPSCs would enhance the manuscript's organization and coherence.

It is also crucial to thoroughly discuss the recent concerns regarding the tumorigenic and tumor-supporting risks associated with MSCs. Addressing these potential safety concerns will provide a comprehensive view of the subject. Moreover, the safety concerns related to the use of iPSCs are not mentioned and should be included.

Among the 31 references cited from the past five years, 21 (68%) are review articles. The novelty of the current manuscript appears to be limited, as it lacks an in-depth discussion by the authors. Therefore, the value of publishing this review may be questionable.

Author Response

Dear Reviewer,

We have tried our best to follow your comments and recommendations. We extensively modified the manuscript and hope the changes meet with your approval. Please consider that the other Reviewers have required other changes, so we have also modified the text at points you did not mention. In the following, we have italicized all your comments and inserted our answers in regular font.

This manuscript provides an interesting summary of the development of cell-based therapies for cartilage regeneration. However, several issues need to be addressed. 

  1. The text includes statements such as, “In 1994, the turning point. For the first time… (line 56), “A cellular approach for the treatment of cartilage defects started in the 1960s… (line 89),” “The results of the first pilot study trials were published in 1994… (line 93),” and “The past three decades have been devoted to regenerative medicine… (line 471).” However, the claim of “Forty years” in the title lacks clarity and precision, which is important for understanding the work.

Thank you for your comment, on which we agree. Indeed, if we consider the first clinical trial, the years are thirty and not forty, while the use of autologous chondrocyte transplantation in animal studies was reported in previous years, as referenced in the text. For this reason, our title was intended to cover all this research. However, since our intention was to highlight the biological research aspects, we modified the text accordingly.

  1. Additionally, the inclusion of induced pluripotent stem cells (iPSCs) in the section on mesenchymal stem cells (MSCs) is not well justified. Creating a separate section for iPSCs would enhance the manuscript's organization and coherence.

Thank you for the opportunity to improve our manuscript. We added a separate section for iPSCs.

  1. It is also crucial to thoroughly discuss the recent concerns regarding the tumorigenic and tumor-supporting risks associated with MSCs. Addressing these potential safety concerns will provide a comprehensive view of the subject. Moreover, the safety concerns related to the use of iPSCs are not mentioned and should be included.

Thank you for the suggestions. We added in the specific paragraphs the relevant concerns related to safety for both MSCs and iPSCs

  1. Among the 31 references cited from the past five years, 21 (68%) are review articles. The novelty of the current manuscript appears to be limited, as it lacks an in-depth discussion by the authors. Therefore, the value of publishing this review may be questionable.

We thank the Reviewer for his/her comment. We are aware that there are many reviews on the same topic. However, the novelty of our work consists in the fact that it intended to review the biological aspects of the different “orthobiological” therapies rather than to review their clinical use, indeed more on the “research” side than on the “clinical” one.  We tried to emphasize this concept in the Introduction.

Reviewer 2 Report

Comments and Suggestions for Authors

The review paper conducted by Grigolo et al. provides a comprehensive review of various therapeutic approaches for cartilage treatment and regeneration. The paper offers an extensive overview of current strategies, from cell-based therapies to emerging cell-derived products. However, while the review is informative, there are several areas where the authors could enhance the paper's clarity and impact:

1-         While the paper offers in-depth information on many topics, some sections (e.g., chondrocytes and MSCs) are significantly more detailed than others (e.g., gene therapy). Consider balancing the level of detail across all sections or explicitly stating why certain topics are given more emphasis

2-         The paper could benefit from more discussion on the clinical translation of these therapies. Consider adding a section or expanding existing content to address the current status of clinical trials, regulatory approvals, and real-world applications of these technologies.

3-         The paper presents various approaches but could be strengthened by including more direct comparisons between different therapies. A table or visual aid summarizing the advantages and limitations of each approach would be valuable for readers.

4-         The discussion touches on future perspectives, but this section could be expanded. Consider elaborating on emerging trends, challenges to be addressed, and potential breakthroughs that could shape the field of cartilage regeneration in the coming years.

5-         The paper could be significantly enhanced by incorporating additional references on cell encapsulation for cartilage repair. For instance, recent studies have explored innovative approaches using injectable hydrogels and advanced biomaterials to encapsulate various cell types for cartilage regeneration. Including such references would provide a more comprehensive overview of current advancements in the field and strengthen the paper's discussion on tissue engineering strategies for cartilage repair: https://link.springer.com/article/10.1007/s10856-011-4396-2

 6-         Incorporating more high-quality figures could significantly enhance the paper's impact and readability. Specifically, adding well-designed figures that summarize key concepts, comparative tables, and infographics would help readers grasp complex ideas more easily and provide quick references to important information.

Author Response

Dear Reviewer,

We have tried our best to follow your comments and recommendations. We extensively modified the manuscript and hope the changes meet with your approval. Please consider that the other Reviewers have required other changes, so we have also modified the text at points you did not mention. In the following, we have italicized all your comments and inserted our answers in regular font.

The review paper conducted by Grigolo et al. provides a comprehensive review of various therapeutic approaches for cartilage treatment and regeneration. The paper offers an extensive overview of current strategies, from cell-based therapies to emerging cell-derived products. However, while the review is informative, there are several areas where the authors could enhance the paper's clarity and impact:

  1. While the paper offers in-depth information on many topics, some sections (e.g., chondrocytes and MSCs) are significantly more detailed than others (e.g., gene therapy). Consider balancing the level of detail across all sections or explicitly stating why certain topics are given more emphasis.

We agree with your comment and added more details about balancing all sections as also requested by another Reviewer.

  1. The paper could benefit from more discussion on the clinical translation of these therapies. Consider adding a section or expanding existing content to address the current status of clinical trials, regulatory approvals, and real-world applications of these technologies.

Thank you for the comment. However, our work was intended to review the biological aspects of the different “orthobiological” therapies rather than their clinical use, indeed more researcher's side than the clinician's. For this reason, we did not add a section on regulatory aspects and current applications that other clinical reviews have already reported in depth.

  1. The paper presents various approaches but could be strengthened by including more direct comparisons between different therapies. A table or visual aid summarizing the advantages and limitations of each approach would be valuable for readers.

Thank you for the comment. For the same reason reported above (n. 2) we did not include a table comparing the different therapeutical options.

  1. The discussion touches on future perspectives, but this section could be expanded. Consider elaborating on emerging trends, challenges to be addressed, and potential breakthroughs that could shape the field of cartilage regeneration in the coming years.

Thank you for the suggestions. We tried to improve the part relative to future perspectives particularly from a biological point of view in the Discussion.

  1. The paper could be significantly enhanced by incorporating additional references on cell encapsulation for cartilage repair. For instance, recent studies have explored innovative approaches using injectable hydrogels and advanced biomaterials to encapsulate various cell types for cartilage regeneration. Including such references would provide a more comprehensive overview of current advancements in the field and strengthen the paper's discussion on tissue engineering strategies for cartilage repair: https://link.springer.com/article/10.1007/s10856-011-4396-2.

We thank the Reviewer and add what suggested in the text including specific references.

  1. Incorporating more high-quality figures could significantly enhance the paper's impact and readability. Specifically, adding well-designed figures that summarize key concepts, comparative tables, and infographics would help readers grasp complex ideas more easily and provide quick references to important information.

We thank the Reviewer and add some new figures.

Reviewer 3 Report

Comments and Suggestions for Authors

The manuscript submitted to Pharmaceutics summarizes the achievements in the field of cell transplantation technique for cartilage regeneration. The textual part of the manuscript is well written. However, the manuscript has several lacks which should be revised before the consideration for publication.  

1. The manuscript is not illustrated at all. The manuscript should be provided with 3-4 summarizing or original figures for better clarity and attractiveness to the readers.

2. A table(s) summarizing the achievements in the submitted work is also highly desirable. 

3. The novelty of this review among other similar reviews (which should also be cited) should be emphasized.

4.  The limitations of the techniques discussed should be clearly stated.

5. The authors should indicate the future outlook for the developments in the discussed medical area.

Author Response

Dear Reviewer,

We have tried our best to follow your comments and recommendations. We extensively modified the manuscript and hope the changes meet with your approval. Please consider that the other Reviewers have required other changes, so we have also modified the text at points you did not mention. In the following, we have italicized all your comments and inserted our answers in regular font.

The manuscript submitted to Pharmaceutics summarizes the achievements in the field of cell transplantation technique for cartilage regeneration. The textual part of the manuscript is well written. However, the manuscript has several lacks which should be revised before the consideration for publication.  

  1. The manuscript is not illustrated at all. The manuscript should be provided with 3-4 summarizing or original figures for better clarity and attractiveness to the readers.

We thank the Reviewer and add some new figures.

  1. A table(s) summarizing the achievements in the submitted work is also highly desirable. 

We thank the Reviewer for his/her suggestion. However, we believe that, in our work, there is too much data to report in a table. Moreover, including such a table would be redundant and would not contribute additional information to the text.

  1. The novelty of this review among other similar reviews (which should also be cited) should be emphasized.

The novelty of our work consists in the fact that it intended to review the biological aspects of the different “orthobiological” therapies than to review their clinical use, indeed more on the “research” side rather than on the “clinical” one. We tried to emphasize this concept in the text.

  1. The limitations of the techniques discussed should be clearly stated.

Thank you for the comment. However, we did not discuss the limitations of the different techniques according to what was explained in the answer to comment 3.

  1. The authors should indicate the future outlook for the developments in the discussed medical area.

Thank you for the suggestions. We tried to improve the part relative to future outlook particularly from a biological point of view in the Discussion.

Round 2

Reviewer 2 Report

Comments and Suggestions for Authors

The authors thoughtful revisions have significantly enhanced the quality and clarity of the paper. The manuscript now presents a more comprehensive and well-rounded discussion of the topic.

Reviewer 3 Report

Comments and Suggestions for Authors

The authors have revised the manuscript according to the reviewer's comments. The manuscript now looks much more attractive and can be recommended for acceptance in its current form.